# Bridging the Methodological Gap Between Inertial Sensors and Optical Motion Capture: Deep Learning as the Path to Accurate Joint Kinematic Modelling Using Inertial Sensors

**DOI:** 10.3390/s25185728

**Published:** 2025-09-14

**Authors:** Vaibhav R. Shah, Philippe C. Dixon

**Affiliations:** 1Institute of Biomedical Engineering, Faculty of Medicine, University of Montreal, Montreal, QC H3T 1J4, Canada; vaibhav.shah@umontreal.ca; 2Centre de Recherche Azrieli du CHU Sainte-Justine (CRCHUSJ), Montreal, QC H3T 1C5, Canada; 3Department of Kinesiology and Physical Activity, McGill University, Montreal, QC H2W 1S4, Canada

**Keywords:** biomech loss function, deep learning, gait, inertial measurement unit (IMU), kinematics prediction, marker prediction, wearable sensors

## Abstract

As advancements in inertial measurement units (IMUs) for motion analysis progress, the inability to directly apply decades of research-based optical motion capture (OMC) methodologies presents a significant challenge. This study aims to bridge this gap by proposing an innovative deep learning approach to predict marker positions from IMU data, allowing traditional OMC-based calculations to estimate joint kinematics. Eighteen participants walked on a treadmill with seven IMUs and retroreflective markers. Trials were divided into normalized gait cycles (101 frames), and an autoencoder network with a custom Biomech loss function was used to predict 16 marker positions from IMU data. The model was validated using the leave-one-subject-out method and assessed using root mean squared error (RMSE). Joint angles in the sagittal plane were calculated using OMC methods, and RMSE was computed with and without alignment using dynamic time warping (DTW). The models were also tested on external datasets. Marker predictions achieved RMSE values of 2–4 cm, enabling joint angle predictions with 4–7° RMSE without alignment and 2–4° RMSE after DTW for sagittal plane joint angles (ankle, knee, hip). Validation using separate and open-source datasets confirmed the model’s generalizability, with similar RMSE values across datasets (4–7° RMSE without DTW and 2–4° with DTW). This study demonstrates the feasibility of applying conventional biomechanical models to IMUs, enabling accurate movement analysis and visualization outside controlled environments. This approach to predicting marker positions helps to bridge the gap between IMUs and OMC systems, enabling decades of research-based biomechanical methodologies to be applied to IMU data.

## 1. Introduction

Joint kinematics are essential biomarkers in physical assessments for rehabilitation, sports science, and clinical applications [1,2,3]. These biomarkers provide insights into movement patterns, enabling the diagnosis and treatment of mobility impairments and optimizing physical performance. Traditionally, calculating joint kinematics has depended on marker-based optical motion capture (OMC) systems, which, while offering high accuracy, are expensive, require significant setup time, and are confined to laboratory environments. These constraints make OMC unsuitable for outdoor monitoring [4,5]. Furthermore, laboratory-based gait analysis may influence natural walking patterns. For instance, Renggli et al. [5] noted that older adults walked faster in lab settings than in real-world environments. Similarly, Carcreff et al. [6] highlighted differences in spatiotemporal gait parameters among children with cerebral palsy across different settings (laboratory vs. daily life), underscoring the need for alternative methods capable of operating effectively outside laboratory constraints.

Advances in wearable technology, particularly inertial measurement units (IMUs), have emerged as promising solutions for continuous, real-world monitoring [3,7,8]. These compact, cost-effective, and portable devices can be easily attached to the body for movement analysis and are reasonably accurate, with 3–6° root mean squared error (RMSE) in joint kinematics [1,9,10]. IMUs can overcome many limitations associated with OMC systems; however, their use in joint kinematic analysis remains challenging. Unlike OMC, which relies on marker positions, IMUs estimate joint kinematics indirectly based on sensor orientation [11,12,13,14,15]. Moreover, IMUs are susceptible to noise, magnetic interference, sensor drift, and gravitational effects, necessitating advanced computational approaches to extract valid and reliable results. Sensor fusion techniques have shown promise, but current algorithms still face challenges (e.g., tuning parameters, dynamic movements, and system noise). Broader adoption is also constrained due to the high cost of software licenses to automate these processes and difficulties in replicating workflows for open science.

Recent research has explored deep learning for predicting joint kinematics using IMUs. Studies suggest that IMU-based deep learning models can achieve RMSE values of 3–8° compared to OMC [16,17,18]; however, these studies often suffer from limitations, such as small datasets [16,19], task-specific models [20,21], or reliance on simulated training data [22,23]. Despite these advancements, significant hurdles still need to be overcome for the practical application of IMU-based deep learning models. For instance, model accuracy can vary depending on gait tasks and speeds, necessitating validation across different activities [22]. Our past research addressed this issue by developing a model that predicted joint kinematics across different speeds (walking, jogging, running) [18]. Nonetheless, two key issues remain in directly predicting joint kinematics from IMU data. First, although joint angles can be used to drive a multi-segment model for visualization, this top-down approach offers limited transparency, making it difficult to identify and interpret the source of errors in the predicted movements. Second, such methods often bypass intermediate representations, such as marker positions or segment orientations, which form the foundation of well-established and validated biomechanical analysis frameworks. To address these challenges, we adopted a bottom-up strategy that enables interpretable, step-by-step validation and facilitates integration with conventional analysis methods familiar to the biomechanics community.

These challenges led us to conduct the present study. Our work aims to develop a deep learning model capable of predicting marker positions directly from IMU data. We validate our models on an open-source dataset where participants walked on a treadmill and overground conditions [24]. We hypothesize that (1) relative marker positions will be accurately predicted from IMU data and (2) using predicted marker positions as inputs to biomechanical models will result in an accurate estimation of sagittal plane lower-limb joint kinematics. Our approach allows researchers to visualize motion data and apply existing biomechanical models developed and validated over the past 40 years on OMC data to new IMU-based gait data inside and outside the laboratory.

## 2. Methods

### 2.1. Participants

Eighteen healthy adults (5 females; age: 25 ± 3 years; height: 170.5 ± 7.9 cm; weight: 66.1 ± 9.3 kg), all with normal limb function and no walking abnormalities, participated in this study. Ethical approval was obtained from the University of Montreal’s ethics committee, and all participants provided written informed consent before data collection (25 May and 30 May 2023).

### 2.2. Data Collection

This study obtained data from IMU sensors (Xsens, Enschede, The Netherlands) and an OMC system (Vicon, Oxford, UK). The ranges of the accelerometer and gyroscope sensors are ±2000 deg/s and ±160 m/s^2^, respectively, sufficient for our application. Each participant was equipped with seven IMUs positioned at the fifth lumbar vertebra, the midpoint of the lateral thigh and shank on both sides and the upper dorsal surfaces of both feet (Figure 1). OMC data captured using the Plug-in Gait marker set [25] and IMU data recorded through MT Manager software 2021.0 (Xsens, Enschede, The Netherlands) were synchronized via a trigger pulse.

A static trial (participants upright and motionless) was collected before performing tasks on a treadmill. The protocol involved 2 min of walking, jogging, running, jogging, and walking in this sequence at their preferred speeds. Herein, only walking data were used.

### 2.3. Data Pre-Processing

Raw OMC and IMU data were combined into a single file. Data were filtered using a fourth-order Butterworth low-pass filter at a 6 Hz cutoff frequency [26,27,28]. Each trial was segmented into single gait cycles and normalized to 100% of the gait cycle (101 frames) using heel-strikes from OMC data. Afterwards, all the marker positions (16 markers) followed the same mathematical transformation (Equation (1)), where first, a factor equivalent to half the distance between each foot in the gait cycle’s starting frame was subtracted (Equation (2)), and 700 was added to all marker positions to ensure positive values, then divided by 100 to convert the marker position to the decimeter scale so that model can learn appropriately without excessively high loss values. These equations are not a statistical transformation (e.g., normalization or standardization). Rather, they are simply numerical shifting and scaling operations to bring all marker values into a positive, low-magnitude range for stable model training. The process does not affect marker geometry or biomechanics and does not distort inter-marker relationships.(1)$$New_{marker}=\frac{Old_{marker}-factor+700}{100}$$(2)$$factor=\frac{\left(LToe+RToe\right)}{2}$$

### 2.4. Sensor Combination

The current study predicts 16 lower-limb marker positions of the Plug-in Gait model using the filtered angular velocity and linear acceleration data from the seven sensors.

### 2.5. Model Architecture

A 4D LSTM-Attention Encoder–Decoder Model architecture was configured with the input size (n, 101, 4, 14) and output (n, 101, 3, 16). The model consisted of multiple LSTMs, multi-head attention and convolution layers (Conv1D). Acceleration and gyroscope data (2) in three axes and their magnitudes (4) for all seven sensors (7) were input into the model. The OMC-acquired marker positions were used as an output to train the prediction model. Model architecture is presented in Appendix A. The model was trained for 200 epochs with a batch size of n = 200, a learning rate of 0.005, an activation function (LSTM: tanh, Conv1D: relu, and Forwardfeed: linear), an Adam optimizer, and a custom loss function (Appendix A). Model training was performed in Python 3.10 using TensorFlow models on Google Collab with an L4 GPU configuration runtime [29]. The architecture and hyperparameters used in this study were adopted mainly from our previous work [18], which involved extensive exploration of deep learning models for IMU-based joint angle prediction. One important modification was the addition of an attention layer to the model architecture. An attention mechanism was introduced due to insufficient prediction accuracy in early testing. This was introduced to better handle the longer sequence lengths in the current dataset (normalized to 101 frames vs. shorter input windows). The attention mechanism enables the model to selectively focus on relevant temporal features across the gait cycle, helping the model understand the continuous nature of motion.

### 2.6. Custom Biomech Loss Function

We opted for the development of a custom loss function (RMSEs for marker position were 4.7 ± 10 cm and 16.5 ± 34.3 deg for joint angles with standard mean square error loss function) (Appendix A, Appendix A). Hence, this study introduces the custom_biomech_loss function, tailored for biomechanical tasks, integrating multiple components to ensure accurate, consistent, and biomechanically plausible predictions. This function has three main components. (1) It combines general mean squared error (MSE) with an extra focus on foot markers, (2) the loss includes a term that prioritizes minimizing the largest prediction errors (top-24 features), ensuring the model addresses significant discrepancies, and (3) an added layer of MSE comparison that aims to restrict prediction to smooth and realistic temporal dynamics, by penalizing abrupt deviations across time steps. The loss function equation is provided in Appendix A and the code for the loss function is available in the Data Availability Statement section. The weights in the custom loss function were determined empirically.

### 2.7. Early-Stopping Configuration

An early-stopping mechanism was implemented to avoid overfitting using TensorFlow’s EarlyStopping callback function [30]. It monitored the validation loss and stopped training if the loss did not improve (at least 100 epochs, patience = 5 epochs, restore_best_weights = True).

### 2.8. Model Evaluation

A leave-one-subject-out approach was used to evaluate model performance, ensuring generalizability to unseen data and avoiding data leakage regarding participant characteristics between training and testing sets [18,27]. To maintain consistency during training, we used Participant 8 data as validation data in the early stopping of the model training for the 17 other participants. Participant 3 data were used as a validation set for training a model for Participant 8. In total, 18 different models were trained. Using the same subject (Participant 8) for early stopping in all model trainings ensures reproducibility and consistency. Changing the validation subject for each training run would introduce variability in the early-stopping condition, potentially biasing model convergence differently across folds. Keeping validation fixed ensures uniform conditions.

For marker positions, RMSE was calculated and averaged across all participants. For joint kinematics, both OMC and predicted marker positions were computed according to the Plug-in Gait model using the BiomechZoo toolbox [31]. Computations require true static data marker positions to extract joint centers. Computed joint kinematics from OMC and predicted marker positions were compared with and without DTW [18]. DTW is a technique used to assess the similarity between two temporal sequences that may vary in speed or length [32,33,34]. DTW improves post-prediction comparison by aligning time-series data and minimizing the gait event timing discrepancies via time-axis warping [18].

### 2.9. External Validation

We validated our models with (a) a new participant (n = 1) collected for a different study in our laboratory, with 2 min of walking on the treadmill and (b) using an open-source database comprising 21 participants [24]. For the latter, the trained model was deployed on overground (n = 11) and treadmill (n = 8) walking trials. In the open-source database, some participants were excluded due to extensive gaps in OMC data, defective IMU data, osteoarthritis-related conditions, and missing treadmill data. Gaps in OMC data from external datasets were filled using the kinetic toolkit in Python [35], and joint kinematics were calculated using a similar approach to Saegner et al. [36]. We used the average prediction of 18 models (1 × 18 participants) to evaluate the models’ performance. This approach may minimize the effect of a single poor-performing model [18]. No cross-scene migration processing or domain adaptation was applied to external data, except for correcting sensor orientations. Instead, the models were evaluated under realistic deployment conditions, where sensor orientations, marker sets, and walking conditions varied.

## 3. Results

### 3.1. Marker Positions

Prediction errors across all markers varied between 2 and 4 cm. As summarized in Table 1, predicting the marker’s Y direction (medio-lateral) was the most challenging, with an RMSE of approximately 4 cm, followed by the Z direction (inferior–superior or height) at 3 cm. The X direction (antero-posterior) was the most accurate, with an error of approximately 2 cm. Figure 2 presents predicted and OMC markers for visualization over a single gait cycle normalized to 100% in 10% increments.

External validation on separately collected data in our laboratory showed similar errors across all three directions: X (1.7 ± 0.5 cm), Y (3.0 ± 0.7 cm), and Z (2.0 ± 0.2 cm). Due to different marker sets used in the external dataset (cluster vs. Plug-in Gait), a direct comparison of marker positions was not conducted. Validation was only conducted on final joint kinematic outputs.

### 3.2. Joint Kinematics

Predicted marker positions accurately calculated gait kinematics using existing biomechanical methodologies. For sagittal plane angles across all three joints, RMSE values ranged between 4 and 7° without aligning prediction data with DTW techniques and 2 to 4° after aligning predicted data with DTW, as shown in Table 2 and Figure 3.

Validation with separately collected data in our lab revealed lower errors across all joints with DTW (1–2° RMSE) and without DTW (2–4° RMSE) (Table 2 and Figure 4). Our model’s performance on the [24] treadmill data showed slightly higher RMSE values than initial validation for the right knee sagittal angles (non-DTW 7.6° and DTW 3.2°) and left ankle sagittal angles (non-DTW 6.4° and DTW 2.8°). Conversely, slightly lower errors were noted for the rest of the joints (<5.4° non-DTW and <2.2 DTW, Table 2, Figure 5) [24]. On the overground walking data of [24], only right hip sagittal angles had slightly lower RMSE (non-DTW 4.6° and 3.4° DTW), while the rest of the joints showed slightly higher errors (<7.2° and 4.7° for non-DTW and DTW, respectively, Table 2; Figure 6).

## 4. Discussion

This study implemented an IMU-based deep learning model to predict marker positions during treadmill walking and used the predicted marker positions to compute joint kinematics. The results demonstrate that the model can predict marker positions with low error, ranging from 2 to 4 cm, with similar performance observed on separately collected and external validation data. Using the predicted marker positions to calculate joint kinematics resulted in an RMSE of less than 7° for initial validation, and comparable performance was noted in separately collected and external validation data. Overall, this study demonstrates the feasibility of a novel approach to computing joint kinematics from IMUs for gait analysis, harnessing modern deep learning algorithms and traditional biomechanical modeling approaches.

### 4.1. Marker Positions

Our first hypothesis was confirmed, as the deep learning model accurately predicted marker positions from IMU data. To the best of our knowledge, this is the first time this approach has been presented in the literature. Thus, we focus on comparing the performance of our model against other approaches that reference OMC as the gold standard. The validation studies of markerless systems such as OpenCap [37] and Theia Markerless [38] report approximately 2–4 cm errors compared to OMC for joint centers and marker positions. We achieve similar performance without inputting subject-specific information such as anthropometric data. Two key factors influenced our model’s performance. The first key factor was aligning OMC markers to a reference starting position and ensuring all marker position values were positive. Second, constructing a custom loss function allowed predictions to abide by certain biomechanical constraints of the human body. Of these, the custom loss function was most impactful in improving accuracy (Appendix A).

### 4.2. OMC Vs. IMU Joint Kinematics

Our second hypothesis was confirmed, as results showed good agreement in computed lower-limb joint kinematics between OMC and joint angles computed with predicted marker positions (RMSE of less than 7°). Secondary analysis also showed strong correlations (correlation coefficient > 0.98 for all joints). The current study’s results showed similar performance compared to recent studies, which used a direct approach to predict joint kinematics from IMU data [16,20,22,39,40,41] (Appendix A). Compared to our previous study [18], which directly predicted joint angles, the presented approach shows slight improvements in hip angle estimations, but similar or marginally worse errors for knee and ankle angles [18]. This difference suggests a trade-off in our current approach: instead of directly predicting a single joint angle, we predict the positions of 16 markers. This makes it challenging for the model to capture extreme motions accurately (e.g., peak knee flexion during swing), which could limit its ability to represent the full range of motion (Table 2). The current method underestimates peak knee flexion and plantar flexion during swing (Appendix A). In practice, this would mean that clinicians may see slightly “flattened” peaks compared to OMC data; however, the overall trajectory shape and timing are preserved, allowing for meaningful interpretation of gait cycle dynamics. Despite the reduced performance, a major advantage of the current approach is the ability to assess face validity via visualization of marker data. Moreover, as errors of less than 5° are acceptable for joint angles in clinical practice [42], our approach is reasonable. Also, as previously mentioned, this approach permits the use of OMC-based joint kinematics calculations such as the conventional gait model (Plug-in Gait) on IMU data [25]. Finally, we present a walking skeleton from a random trial for all participants tracked via predicted marker positions in Appendix A and the Data Availability Statement section to further support our findings.

### 4.3. External Validation and Generalizability

External validation on separately collected and open-source data confirmed the model’s generalizability. Prediction errors for separately collected data aligned with the model’s initial validations, and we also found consistent RMSEs across joint kinematics. Ankle joint predictions were slightly worse than for the knee and hip. This may be expected as predicting three non-linear markers on the foot and ankle complex with one sensor can be difficult for the model. For external validation, we tested our model on a treadmill and with overground preferred-speed walking from a public dataset [24]. For treadmill walking, overall results were slightly better for joints other than the right knee and right ankle, while for overground walking, only right hip angles were slightly better. At the same time, errors were slightly worse than initial validation for knee and ankle joint angles, while left hip angles showed similar errors. Two key issues in the angle predictions were observed. First, the model could not appropriately predict knee flexion during the swing phase, which led to lower flexion estimates. Second, during the swing phase, the model also had difficulty predicting plantar flexion of the ankle joint.

Several factors in the external dataset affected the model’s ability to accurately predict marker positions, leading to higher errors than initial validation and our past study [18]. First, the sensor on the shank was positioned more distally (closer to the ankle) than in our dataset. Second, sensor orientations differed. Only the pelvis, right thigh, and right shank sensors were in the same orientation as ours. The sensor data of both feet showed a rotation of 180° around the sensor’s Z-axis, and the left thigh and shank sensor data were rotated 180° around the sensor’s X-axis. Rotating sensor data corrected orientations but may have introduced minor differences in the data that confused the model. Third, external data had different marker sets (cluster vs. Plug-in Gait), requiring a different approach to calculating joint kinematics for OMC data. Despite these challenges, the model provided a good overall agreement between OMC and IMU joint angles (Figure 5 and Figure 6). We also tested the robustness of our model by interchanging the locations of the IMU sensors to ensure that it does not produce similar predictions regardless of input data. Specifically, we swapped the foot sensor data with the shank sensor data and vice versa. This test confirmed that the model does not generate gait cycle-like predictions for arbitrary input configurations, demonstrating its ability to discern meaningful patterns based on the correct sensor placement (see Appendix A).

### 4.4. Conclusion, Limitation and Outlook

This study demonstrates the ability of IMU-based deep learning models to accurately predict marker positions for use in lower-limb joint kinematic computations for gait analysis. The proposed approach successfully bridges traditional marker-based methodologies and wearable IMU technology by leveraging predicted marker positions, achieving results comparable to other alternative approaches to OMC systems. The findings highlight an innovative path for motion analysis beyond controlled laboratory environments, opening possibilities for applications in rehabilitation, sports science, and clinical assessment.

Several limitations warrant attention. First, the model was trained exclusively on typical walking patterns, and its performance on other gaits (e.g., jogging, running) or pathological movements remains unknown. The method is currently validated for treadmill sagittal plane gait in healthy adults. Additionally, this study focused solely on sagittal plane kinematics, limiting its scope for comprehensive gait analysis across all planes. Moreover, there is a need to address sensor positioning variability and expand the dataset to include diverse sensor placements and motion conditions. Additionally, the custom Biomech loss function is currently limited to the walking task, and further testing is required for non-walking tasks.

Future research should address these limitations by incorporating more diverse datasets, including pathological gaits and daily activities. Expanding the model to predict kinematics across all three planes with proven OMC methodology will enhance its utility for complex motion analysis. Additionally, integrating user-specific data, such as anthropometric information, could improve the accuracy and generalizability of the predictions. Efforts to validate the approach in real-world environments, such as home-based rehabilitation or sports performance monitoring, will be critical for its broader adoption.

## 5. Conclusions

In conclusion, this study presents a novel approach for IMU-based kinematic analysis, combining the strengths of traditional biomechanical computational models with modern wearable technologies. Moreover, this approach allows for visualization and analysis of human motion outside laboratory settings. This method lays the groundwork for more accessible, real-world applications of biomechanical research, ultimately advancing fields such as rehabilitation, clinical diagnostics, and athletic performance.

## Figures and Tables

**Figure 1 sensors-25-05728-f001:**
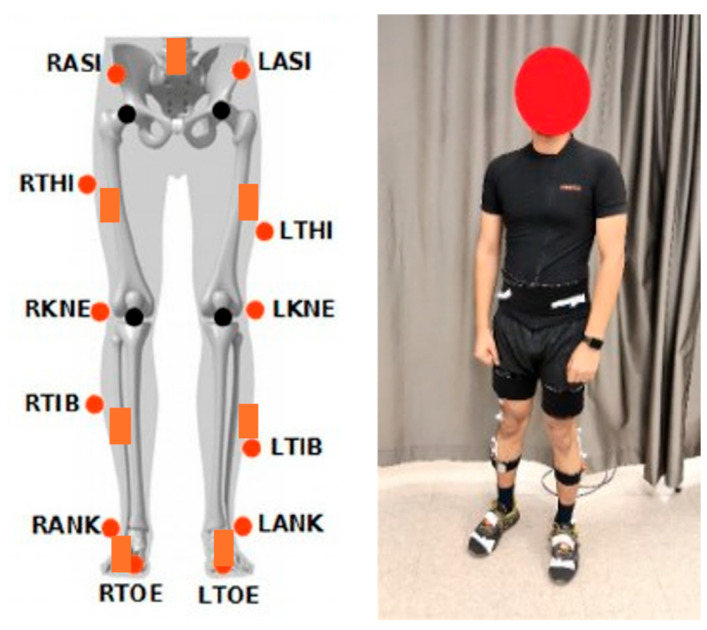
Sensor and marker position. (IMU sensors in orange) c.f. [18].

**Figure 2 sensors-25-05728-f002:**
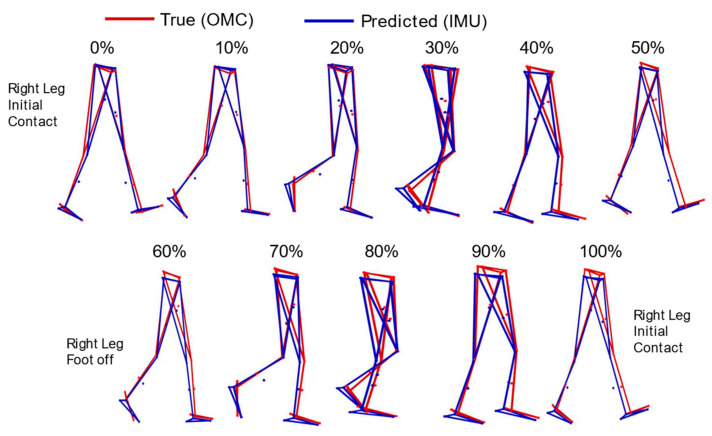
Visual representation of a random walking stick figure from 0 to 100% of the gait cycle constructed from optical motion capture (OMC) (red) and inertial measurement unit predicted (blue) marker positions (small dots) at every 10% step of the gait cycle.

**Figure 3 sensors-25-05728-f003:**
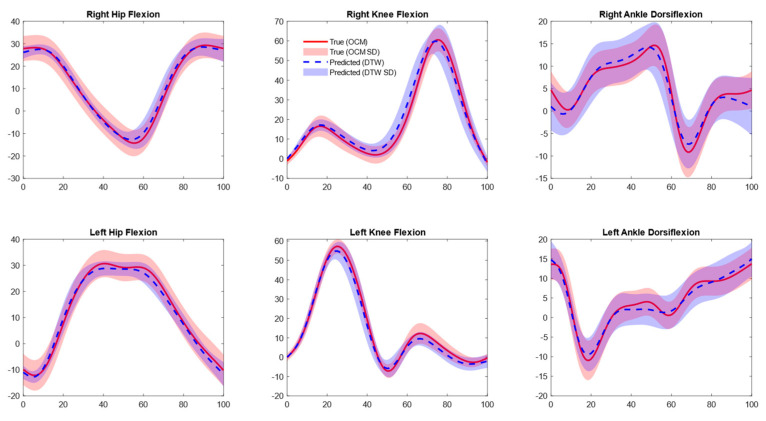
Mean joint angle plots for optical motion capture (OMC) vs. inertial measurement unit (IMU)-based approach. Predictions shown are aligned with OMC data using dynamic time warping (DTW).

**Figure 4 sensors-25-05728-f004:**
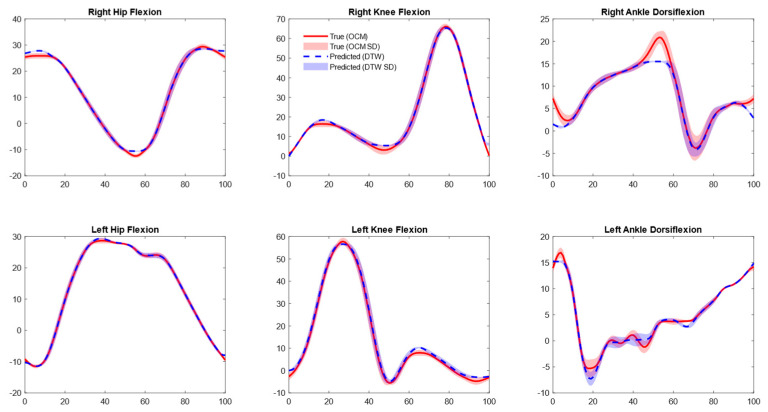
Mean joint angle plots for optical motion capture (OMC) (Separately collected data) vs. inertial measurement unit (IMU)-based approach. Predictions shown are aligned with OMC data using dynamic time warping (DTW).

**Figure 5 sensors-25-05728-f005:**
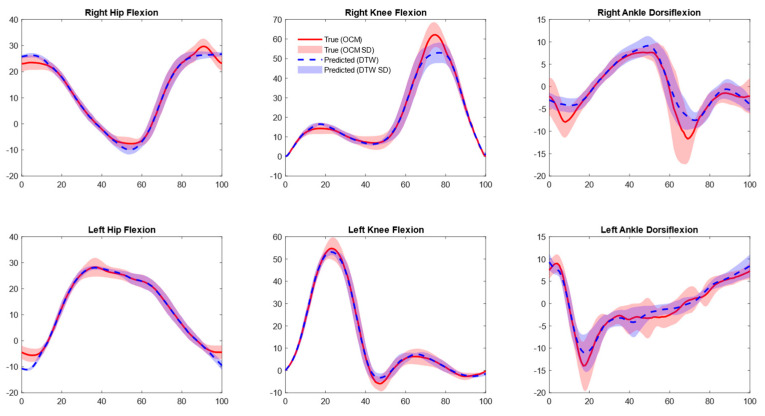
Mean joint angle plots for optical motion capture (OMC) (external treadmill walking data [24]) vs. inertial measurement unit (IMU)-based approach. Predictions shown are aligned with OMC data using dynamic time warping (DTW).

**Figure 6 sensors-25-05728-f006:**
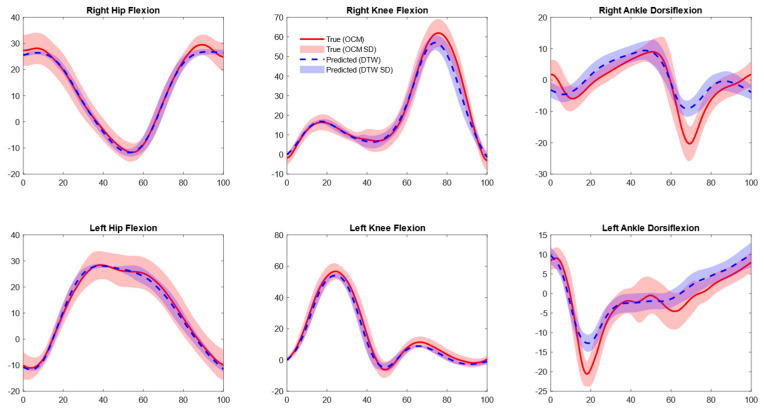
Mean joint angle plots for optical motion capture (OMC) (external treadmill overground walking data [24]) vs. inertial measurement unit (IMU)-based approach. Predictions shown are aligned with OMC data using dynamic time warping (DTW).

**Table 1 sensors-25-05728-t001:** Average prediction error of predicted marker positions against optical motion capture (OMC) marker positions at initial validation in our data presented in centimeters (cm).

Definition	Markers	X (cm)	Y (cm)	Z (cm)
Left ASIS	LASI	2.3 ± 0.8	2.4 ± 0.9	3.8 ± 2.0
Right ASIS	RASI	2.1 ± 0.9	2.6 ± 0.8	3.6 ± 1.8
Left PSIS	LPSI	2.3 ± 0.5	3.0 ± 1.5	3.1 ± 1.6
Right PSIS	RPSI	2.3 ± 0.7	3.1 ± 1.4	3.2 ± 1.7
Left thigh	LTHI	2.3 ± 0.6	3.2 ± 1.5	4.6 ± 2.7
Left knee	LKNE	2.9 ± 1.2	3.4 ± 1.2	3.0 ± 1.6
Left tibia	LTIB	2.7 ± 0.9	4.0 ± 1.8	3.3 ± 2.2
Left ankle	LANK	3.0 ± 1.4	4.3 ± 2.7	2.3 ± 0.8
Left heel	LHEE	2.5 ± 0.6	4.2 ± 2.5	2.4 ± 0.9
Left toe	LTOE	2.9 ± 1.0	4.4 ± 3.0	1.6 ± 0.7
Right thigh	RTHI	2.4 ± 0.8	3.8 ± 1.1	5.9 ± 3.4
Right knee	RKNE	2.2 ± 0.6	3.5 ± 1.5	2.7 ± 1.7
Right tibia	RTIB	2.1 ± 0.6	4.4 ± 2.4	3.1 ± 1.8
Right ankle	RANK	2.2 ± 0.5	4.5 ± 2.9	2.0 ± 1.0
Right heel	RHEE	2.6 ± 1.1	4.6 ± 3.1	2.3 ± 1.1
Right toe	RTOE	2.4 ± 0.9	4.9 ± 3.3	1.5 ± 0.5
Average All Markers		2.4 ± 0.8	3.8 ± 2.0	3.0 ± 1.6

Notes: ASIS = Anterior superior iliac spine, PSIS = Posterior superior iliac spine.

**Table 2 sensors-25-05728-t002:** Joint angle comparison based on optical motion capture (OMC) and predicted marker position using our proposed method.

Datasets	Joints	Right Hip	Right Knee	Right Ankle	Left Hip	Left Knee	Left Ankle
Same dataset	Non-DTW	5.2 ± 1.4	5.8 ± 1.3	6.8 ± 2.5	4.5 ± 1.7	5.3 ± 2.2	5.4 ± 2.1
	DTW	2.8 ± 1.2	2.3 ± 0.8	4.0 ± 1.9	1.8 ± 0.7	2.6 ± 1.1	2.1 ± 1.0
Collected separately	Non-DTW	3.3 ± 0.6	3.6 ± 0.7	4.2 ± 0.6	2.8 ± 0.5	3.2 ± 0.5	4.2 ± 0.5
	DTW	1.0 ± 0.2	1.4 ± 0.4	2.1 ± 0.4	0.5 ± 0.1	1.1 ± 0.2	0.8 ± 0.2
External Treadmill [24]	Non-DTW	4.5 ± 1.5	7.6 ± 2.2	6.4 ± 1.9	4.0 ± 1.2	5.0 ± 1.4	5.4 ± 1.9
	DTW	2.1 ± 0.7	3.2 ± 1.1	2.8 ± 1.4	2.1 ± 0.8	2.2 ± 0.8	2.0 ± 0.8
External Overground [24]	Non-DTW	4.6 ± 2.2	7.2 ± 2.2	7.1 ± 2.2	6.0 ± 2.3	5.5 ± 1.9	6.0 ± 1.9
	DTW	3.4 ± 1.2	4.7 ± 2.5	4.6 ± 2.4	4.1 ± 1.9	3.7 ± 1.8	3.9 ± 2.0

All values represent root mean squared error (RMSE) in degrees. DTW = Dynamic time warping was used to align the data, Non-DTW = no alignment of data. All values in deg.

## Data Availability

We provide code and trained models that use one participant’s data from the open-source dataset walking over the ground as an example to test our developed model: https://github.com/Vaibhavshahvr7/Marker-position-prediction (accessed on 10 August 2025). In the same repository, we also provide the code of the loss function: https://github.com/Vaibhavshahvr7/Marker-position-prediction/blob/main/custom_biomech_loss.py (accessed on 10 August 2025).

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
