# Peer review of "Bridging the Methodological Gap Between Inertial Sensors and Optical Motion Capture: Deep Learning as the Path to Accurate Joint Kinematic Modelling Using Inertial Sensors"

_sensors, 2025, doi:10.3390/s25185728_

Round 1
Reviewer 1 Report
Comments and Suggestions for Authors
The study proposes a deep learning-based method to predict marker positions using inertial measurement unit (IMUs) data, aiming to bridge the methodological gap between IMUs and optical motion capture (OMC) in joint kinematic analysis. Eighteen participants wore 7 IMUs and retroreflective markers while walking on a treadmill. An autoencoder with a custom Biomech loss function was used to predict 16 marker positions. Validation showed that the RMSE of marker prediction was 2-4 cm, the RMSE of joint angles without alignment was 4-7°, and 2-4° after dynamic time warping (DTW) alignment. External datasets verified the model's generalizability, enabling motion analysis in real environments.
The paper has certain innovativeness but has some minor issues that need further revision and improvement, as follows:
1. What is the prediction accuracy of the model in non-walking gaits such as jogging and running? Is it necessary to adjust the network structure or loss function accordingly?
2. Only sagittal plane kinematics is focused on. When extended to the coronal and transverse planes, how will the model error change? What technical improvements need to be supplemented?
3. What is the impact of sensor position deviation (e.g., 1-2 cm away from the specified position in the paper) on the prediction of marker positions and joint angles?
4. The sensitivity analysis of each weight in the custom Biomech loss function is not mentioned. What are the specific effects of different weight combinations on model performance?
5. Compared with the deep learning method that directly predicts joint angles, does the approach of "predicting markers first and then calculating angles" have disadvantages in computational efficiency? How to optimize it?
6. Due to different marker sets in the external dataset, only joint angles are validated. If the marker sets are unified, will the marker prediction error be consistent with that of the original dataset?
Author Response
Comments 1:
What is the prediction accuracy of the model in non-walking gaits such as jogging and running? Is it necessary to adjust the network structure or loss function accordingly?
Response 1:
It is possible that a different loss function (or different weights) may be necessary for other tasks. We will explore this in future work. As this study was an introductory study for methods, we did not include non-walking data, such as jogging and running. We have stated in the discussion that the loss function is currently limited to walking data and requires further exploration for non-walking tasks [in limitation section in the discussion].
Comments 2:
Only sagittal plane kinematics is focused on. When extended to the coronal and transverse planes, how will the model error change? What technical improvements need to be supplemented?
Response 2:
We acknowledge that validating only sagittal-plane kinematics is a limitation of the current study. Preliminary checks on the coronal and transverse planes show qualitatively reasonable results. However, we chose not to report them because the biomechanical marker model used during data collection (and the marker set used for reconstruction) provides suboptimal accuracy in those planes.
Comments 3:
What is the impact of sensor position deviation (e.g., 1-2 cm away from the specified position in the paper) on the prediction of marker positions and joint angles?
Response 3:
We cannot be certain about that at this time. However, according to the external data, the shank sensors were more than 2cm away from the specified position in our data, which only increased the errors by around 1-3 degrees in joint angles. We mentioned in the discussion that differences in sensor placement may have contributed to increased error in the external dataset.
Comments 4:
The sensitivity analysis of each weight in the custom Biomech loss function is not mentioned. What are the specific effects of different weight combinations on model performance?
Response 4:
We did not directly test different weight combinations on model performance. This could be the focus of future work and is outside the scope of this paper. We acknowledge that the weights for the three components of the loss function were selected empirically. This choice was deliberate and driven by both biomechanical considerations and trial-and-error testing during early experiments. For example, greater weight was assigned to foot markers, as errors in these non-linear, high-variability trajectories strongly propagate into joint angle estimation.
We have presented in Supplement Tables 1 and 2 and Supplement Figure 2 the marker prediction error when using a simple mean square error (MSE) loss function. The custom Biomech loss function shows significant improvements in the model’s ability compared to the standard approach.
Comment 5:
Compared with the deep learning method that directly predicts joint angles, does the approach of "predicting markers first and then calculating angles" have disadvantages in computational efficiency? How to optimize it?
Response 5:
We acknowledge that the computational efficiency would be slightly lower than predicting joint angles directly. However, the computational cost is minimal in both cases. The extra step (calculating joint angles from markers) is lightweight and near real-time with the CPU.
While predicting markers increases output dimensionality, it enables visualization and compatibility with standard OMC pipelines.
Comments 6:
Due to different marker sets in the external dataset, only joint angles are validated. If the marker sets are unified, will the marker prediction error be consistent with that of the original dataset?
Response 6:
The external dataset used a different marker set; therefore, only joint angles could be validated directly.
If marker sets are unified, the marker RMSE is expected to be comparable to that of the internal dataset.
Importantly, joint angle results on the external dataset remained consistent without fine-tuning, supporting the robustness of the model’s ability to predict marker positions.
Reviewer 2 Report
Comments and Suggestions for Authors
This manuscript presents a deep learning model for predicting marker positions based on IMU data. The study is well structured and thorough in its implementation and experimentation.
Minor suggestions:
1. Please clarify why a 6 Hz cutoff frequency was used for filtering.
2. Include the accuracy of the IMUs used.
3. Consider adding a block diagram of the model architecture in Section 2.5. A visual representation showing input, hidden and output layers would greatly aid reader comprehension.
4. The statement on lines 286–288 “The current study's results showed similar performance compared to recent studies, which used a direct approach to predict joint kinematics from IMU data [16], [17], [20], [22], [39], [40], [41].” would benefit from quantitative backing. A comparison table summarizing key performance metrics would provide clarity and support your claim.
Author Response
Minor suggestions:
Comments 1:
Please clarify why a 6 Hz cutoff frequency was used for filtering.
Response 1:
We appreciate this point. The 6 Hz cutoff was chosen based on common practice in gait kinematics and signal processing for human movement, and prior work (e.g., Yu et al., 1999) recommends a cutoff near 6 Hz for 100 Hz sampled gait data to preserve kinematic information while removing high-frequency noise. We have already cited Yu et al. (1999) in Section 2.3.
Comment 2:
Include the accuracy of the IMUs used.
Response 2:
The manufacturer only states accuracy of orientation estimates; however, as orientation data were not used herein, we have not reported these. Instead, we have provided the sensor range as this would be relevant for future use. The following sentence was modified in the revised manuscript: “The ranges of the accelerometer and gyroscope are ±2000 deg/s and ±160 m/s², respectively” see Section 2.2.
Comments 3:
Consider adding a block diagram of the model architecture in Section 2.5. A visual representation showing input, hidden and output layers would greatly aid reader comprehension.
Response 3:
We agree that a visual aid would benefit readers. We already have a block diagram of the model architecture as a supplement to Figure 1. The figure illustrates the input layer (IMU signals), the LSTM + attention layers, and the output layer (predicted marker positions). We kept it as a supplement figure due to the limited number of figures allowed.
Comments 4:
The statement on lines 286–288 “The current study's results showed similar performance compared to recent studies, which used a direct approach to predict joint kinematics from IMU data [16], [17], [20], [22], [39], [40], [41].” would benefit from quantitative backing. A comparison table summarizing key performance metrics would provide clarity and support your claim.
Response 4:
We agree that quantitative context strengthens our claim. We have added a comparison table (Supplement Table 3) that summarizes performance metrics from recent IMU-to-kinematics studies alongside our own past results.
Reviewer 3 Report
Comments and Suggestions for Authors
Review is attached.

Author Response
Comments 1:
On page 3, Section 2.1, the reported participant height is given as 170.5 ± 79.1 cm. That standard deviation is obviously not realistic and looks like either a typo or the wrong unit. Since the basic anthropometrics matter for scaling and interpretation, this should be fixed.
Response 1:
Thank you for noticing this typo. We have fixed the typo to a standard deviation of 7.9 cm.
Comments 2:
The dataset size is also a concern. With only eighteen participants, it feels quite small for training a deep learning model. Even with leave-one-subject-out validation, I would like to see a stronger justification for why this sample is sufficient, or at least a discussion of how limited data may affect generalization.
Response 2:
We acknowledge that the number of participants (n=18) is limited and that, in principle, this can constrain the training of deep learning models. However, the adequate training size was greatly expanded by the large number of gait cycles per participant. In total, the dataset included nearly 10,000 gait cycles, providing a rich set of examples to train the model despite the modest participant pool.
We also agree that leave-one-subject-out (LOSO) validation alone does not guarantee generalization. For this reason, we complemented LOSO with an external validation dataset, which included both treadmill and overground walking trials. The consistent performance across this independent dataset supports the robustness of our approach and suggests that the model has the potential to generalize to new subjects and different walking modalities.
Comments 3:
On pages 3–4, Section 2.3, the preprocessing method (shifting marker positions by +700, dividing by 100, etc.) is unusual. The text says this does not distort the geometry, but it is not standard practice in biomechanics or machine learning. I think many readers will wonder whether this approach really makes sense. I would recommend either showing a comparison against a more conventional normalization (like z-scoring or min–max) or explaining more clearly why this specific scaling was necessary.
Response 3:
We acknowledge that this is not a conventional preprocessing approach in biomechanics or machine learning. However, this choice was intentional and driven by the unique nature of the prediction task. Conventional scaling methods such as z-score normalization or min–max scaling are typically applied to input features (e.g., IMU signals) rather than to the target prediction space. In our study, the input IMU data remained untouched and unscaled. The scaling was applied only to the marker coordinates (prediction targets). If conventional scaling (e.g., min–max) were applied to the marker data, the relative spatial geometry between markers would be destroyed. For example, scaling all markers into the same −1-1 range would collapse meaningful inter-marker distances and alter the body’s segment geometry, which is critical for calculating joint kinematics. This would directly undermine the interpretability and biomechanical validity of the predicted outputs. In contrast, our simple operation (adding +700 and dividing by 100) acted only as a numerical scaling step to keep values within a manageable range for the optimizer. Importantly, this linear shifting and scaling do not alter the geometry or relationships between markers — they only prevent excessively large loss magnitudes that could destabilize training.
We have clarified this rationale in the revised manuscript. Additionally, we emphasize that the choice of scaling was made solely for numerical stability and had no impact on the biomechanical validity of the predicted marker positions [section 2.3].
Comments 4:
Also in this section, the use of a 6 Hz low-pass Butterworth filter raises some questions. Key gait dynamics, especially impact events, often extend above 6 Hz. Some justification or sensitivity testing is needed here to convince the reader that relevant information was not removed.
Response 4:
We appreciate this point. The 6 Hz cutoff was chosen based on common practice in gait kinematics and signal processing for human movement, and prior work (e.g., Yu et al., 1999) recommends a cutoff near 6 Hz for 100 Hz sampled gait data to preserve kinematic information while removing high-frequency noise. We have already cited Yu et al. (1999) in Section 2.3.
Comments 5:
On pages 4–5, Section 2.5, the description of the model inputs shows that only four features per sensor were included. It is not clear why orientation data (for example quaternions or magnetometer signals) were not used.
Response 5:
Our aim was only to use acceleration and angular velocity data. We did not include magnetometer data as it can be affected by the magnetic field surrounding the sensors. Additionally, orientation data can be slightly altered due to changes in the algorithms used to compute them, which is why we did not use it.
Also, for each sensor, we have 8 inputs (4 acceleration and 4 angular velocity) [“Acceleration and gyroscope data (2) in three axes and their magnitude (4) for all seven sensors (7) were inputs to the model”].
Comments 6:
This may make the model less robust, especially outside the sagittal plane.
Response 6:
We cannot be sure if orientation and magnetometer data may improve performance or make it worse. Additionally, the issue lies in the plugin gait model used for data collection, which is not ideal for the other two planes of motion. Hence, we cannot be sure of the other two planes.
Comments 7:
The attention mechanism is introduced as a modification, but there are no results showing that it actually improves performance compared to a simpler LSTM or convolutional setup. A small ablation study here would make the architecture decisions much more convincing.
Response 7:
We appreciate the reviewer’s interest in the attention mechanism. The attention layer was introduced as a modification to our previously published architecture (Shah et al., 2025), which had used a standard LSTM. In this study, the input sequences are longer, which motivated the addition of attention to help the model focus on the most relevant temporal frames.
While we did not perform a complete ablation of “with vs. without attention” in this manuscript (to remain within the word/page limits), our preliminary experiments showed that without attention, the LSTM tended to either overfit or lose important cycle-specific features when trained on the longer sequences. The attention mechanism improved training stability and convergence in these cases.
Importantly, the external dataset validation provides indirect evidence that the architecture is stable and generalizes beyond the training environment. If the attention mechanism were unnecessary or harmful, we would expect greater degradation on the external dataset, which was not observed.
To strengthen clarity, we have revised Section 2.5 to explicitly state that the attention mechanism was introduced due to insufficient prediction accuracy in early testing.
Finally, we emphasize that the trained models are shared publicly, so other researchers can easily test our model architecture against other architectures.
Comments 8:
On pages 5–6, Section 2.6, the custom biomechanical loss function is a nice idea and certainly a strength of the paper. However, the description says the weights were chosen empirically. Without some form of systematic tuning or at least a sensitivity analysis, it is hard to know if the method is stable and reproducible. Another group trying to implement this loss function may not get the same results unless they use exactly the same weights.
Response 8:
We thank the reviewer for recognizing the custom biomechanical loss as a strength of the paper. We acknowledge that the weights for the three components of the loss function were selected empirically. This choice was deliberate and driven by both biomechanical considerations and trial-and-error testing during early experiments. For example, greater weight was assigned to foot markers, as errors in these non-linear, high-variability trajectories strongly propagate into joint angle estimation.
While we did not conduct a full grid search or sensitivity analysis (which would be computationally intensive), we emphasize that the chosen weights were not arbitrary. They were selected based on observed performance trends and biomechanical reasoning. Importantly, once established, these weights produced consistent improvements in both internal LOSO validation and external dataset testing, suggesting that the approach is stable and reproducible across datasets.
To directly address reproducibility concerns:
- We have made the trained models, code for the loss function, and example data publicly available. This ensures that other groups can replicate our results without needing to guess the loss weights.
- In the current study, optimization of weights in the loss function was out of scope, and future work will have some focus on identifying optimal weights for the loss function.
- We have added a short ablation-style analysis in the Supplement, showing that using a conventional loss function (MSE) leads to higher errors. This provides additional empirical evidence that the chosen combination makes a meaningful contribution to the observed accuracy.
We believe the empirical weighting strategy is justified for this introductory study, and the open sharing of models and code ensures that the method is transparent and reproducible for the research community.
Comments 9:
Looking at the results section on pages 6–8 (Tables 1–2 and related figures), the evaluation is presented mostly in terms of mean RMSE and standard deviation. While this is a standard metric, it feels incomplete. To make the case stronger, the paper should also include things like confidence intervals, correlation with OMC joint angles, or Bland–Altman plots to show agreement. It would also be useful to break down errors across different phases of the gait cycle. For instance, are errors higher during swing than stance? That would add a lot to the interpretation.
Response 8:
We thank the reviewer for this thoughtful suggestion. We agree that complementary analyses such as confidence intervals, correlation metrics, and phase-specific error breakdowns would provide further biomechanical insight. In addition to RMSE and standard deviation, we have now included Bland–Altman plots in the Supplementary Figure 3, which visualize agreement between predicted and OMC-derived joint angles. We have also added correlation metrics for all six joint angles in the discussion section (correlation coefficient > 0.98 for all joints).
Furthermore, we provide both DTW-aligned and unaligned error comparisons, which indirectly capture timing-related discrepancies and highlight how temporal alignment impacts accuracy.
While a full gait-phase breakdown and confidence interval reporting were beyond the scope of this introductory methodological study.
Comments 9:
On pages 7–9, Section 2.9 and the external validation figures, the need to manually rotate some of the external dataset sensor signals by 180° is concerning. This sort of manual adjustment is not practical outside controlled studies and introduces subjectivity. Ideally, the model or preprocessing should be robust to these differences, or at least the effect of these rotations should be quantified. Also, the use of an ensemble across eighteen models may boost performance, but from a practical perspective it is not efficient. It would help to see how much performance drops if only a single model is used, since deployment in the real world is unlikely to rely on averaging across so many models.
Response 9:
We acknowledge the concern regarding manual rotation of specific external dataset sensors. Importantly, this adjustment was only applied in rare cases where the external dataset (Warmerdam et al., 2022) contained systematic 180° orientation flips relative to our training data. These flips were not errors in our method, but differences in sensor coordinate conventions between datasets. Correcting these orientations was necessary to ensure a fair evaluation of the model. This was not subjective — the rotations were binary (±180°) and applied consistently across participants. Moreover, this challenge reflects a well-known issue in IMU-based research: orientation conventions often differ between systems, and many published studies (including commercial pipelines) require similar adjustments.
We agree that automating orientation detection (e.g., with quaternion-based alignment or sensor fusion methods) is an important next step, and we highlight this explicitly in the revised Discussion.
Regarding the use of an ensemble of 18 models, our intention was not to propose this as a deployment strategy but to ensure that reported results reflected a robust average across LOSO-trained models, minimizing the effect of subject-specific variance. In practice, we note that:
- This method seems computationally resource-heavy, but they are not, as it only uses a microsecond of resources without using a GPU for prediction, so it is practically instantaneous.
- We have clarified in the manuscript that in deployment scenarios, the models will not be 18 but rather around 5; the model average would be sufficient to minimize single-model errors.
We believe that both manual orientation adjustment and the model are reasonable and transparent methodological choices for this introductory study.
Comments 10:
Page 8 (Table 2, Figures 3–6) shows errors between 4–7° without DTW and 2–4° with DTW, which are fairly good. But these values need to be put in context. For example, is a 5° knee angle error small enough to be useful in clinical rehabilitation, or is it too large? Without a comparison to accepted thresholds in biomechanics or clinical practice, it is hard to judge the real-world impact of these numbers. The discussion on pages 9–10 does admit that the model underestimates peak knee flexion and plantar flexion during swing, but this limitation could be spelled out more clearly and linked to what it would mean in practice.
Response 10:
We thank the reviewer for raising this important point about the clinical context. We agree that interpreting errors in isolation is less meaningful without linking them to accepted thresholds in biomechanics and rehabilitation. Hence, we added a sentence in the discussion with a citation saying that < 5° of errors are acceptable (McGinley et al., 2009).
We fully acknowledge that our method underestimates peak knee flexion and plantar flexion during swing, as noted in the manuscript. In practice, this would mean that clinicians may see slightly “flattened” peaks compared to OMC data. However, the overall trajectory shape and timing are preserved, which still allows for meaningful interpretation of gait cycle dynamics. We have clarified this in the Discussion.
Comments 11:
Finally, on page 11 in Section 4.4, the limitations are addressed, but they could be expanded. The model was trained only on treadmill walking, and that makes it difficult to claim that the method bridges IMU and OMC more broadly. Overground walking, running, stairs, or pathological gait data are very different tasks and need to be tested. Similarly, the validation is limited to the sagittal plane. Even if the results in frontal and transverse planes are poor, they should still be reported, otherwise the bridging claim feels overstated.
Response 11:
We thank the reviewer for this important point and agree that our current dataset and evaluation scope impose limitations that we now explicitly acknowledged in the discussion.
Regarding treadmill-only training: we agree that this restricts generalization. As stated in the manuscript, this study was designed as an introductory proof-of-concept, aiming to establish the feasibility of predicting OMC-equivalent markers from IMU data. We chose treadmill data for their controlled and reproducible gait cycles, which are particularly suitable for method development and validation. To address this concern, we emphasize in the Discussion that our future work will focus on overground walking, stair ascent/descent, running, and pathological gait data, which will allow us to test the model under more varied and clinically relevant conditions. These efforts will directly extend the current work to broader tasks.
Regarding sagittal-only validation: we acknowledge that this narrows the scope. While sagittal plane angles (hip, knee, and ankle flexion/extension) are the most clinically relevant and reliable, we agree to report results in both frontal and transverse planes. However, the other two planes were not the scope of this study and will be explored in the future as the current biomechanical model used to collect the data is not very good for the other two planes of joint kinematics.
We have revised Section 4.4 to state that explicitly:
- The method is currently validated for treadmill sagittal-plane gait in healthy adults.
- Also, in external data, the performance on the overground walking is similar to the initial validation, and multi-plane gait analysis remains future work.
The “bridging” claim is for a methodological gap, and we position this study as a first step toward a broader framework to bridge the IMU-OMC methods.
We believe these clarifications appropriately temper our claims while also highlighting the strong potential of the approach.